# Rho-Family Small GTPases: From Highly Polarized Sensory Neurons to Cancer Cells

**DOI:** 10.3390/cells8020092

**Published:** 2019-01-28

**Authors:** Takehiko Ueyama

**Affiliations:** Laboratory of Molecular Pharmacology, Biosignal Research Center, Kobe University, Kobe 657-8501, Japan; tueyama@kobe-u.ac.jp

**Keywords:** CDC42, congenital (hereditary) diseases, DFNA1, DIA1 (DIAPH1), GSPT1 (eRF3a), hearing, NADPH oxidase (Nox), RAC, RHOA, reactive oxygen species (ROS)

## Abstract

The small GTPases of the Rho-family (Rho-family GTPases) have various physiological functions, including cytoskeletal regulation, cell polarity establishment, cell proliferation and motility, transcription, reactive oxygen species (ROS) production, and tumorigenesis. A relatively large number of downstream targets of Rho-family GTPases have been reported for in vitro studies. However, only a small number of signal pathways have been established at the in vivo level. Cumulative evidence for the functions of Rho-family GTPases has been reported for in vivo studies using genetically engineered mouse models. It was based on different cell- and tissue-specific conditional genes targeting mice. In this review, we introduce recent advances in in vivo studies, including human patient trials on Rho-family GTPases, focusing on highly polarized sensory organs, such as the cochlea, which is the primary hearing organ, host defenses involving reactive oxygen species (ROS) production, and tumorigenesis (especially associated with RAC, novel RAC1-GSPT1 signaling, RHOA, and RHOBTB2).

## 1. Introduction

The small GTPases (also known as small G proteins (~21 kDa)) of the Rho-family (Rho-family GTPases) form a subfamily of the Ras superfamily of small GTPases. Human and murine Rho-family GTPases include 21 members subdivided into eight families classified as typical (classical) or atypical depending on their mode of regulation [1] (Table 1). Rho, Rac, Cdc42, and RhoF/RhoD are members of the classical subfamilies (Table 1) and their functions are regulated by cycling between an inactive GDP-bound form and an active GTP-bound form. Rnd, RhoH, RhoU/RhoV, and RhoBTB are atypical subfamilies and predominantly exist in the GTP-bound-form. The GTP-bound status is regulated by guanine nucleotide exchange factors (GEFs), GTPase-activating proteins (GAPs), and Rho-specific guanine nucleotide dissociation inhibitors (RhoGDIs) [2,3]. In humans, there are ~80 RhoGEFs classified into the Dbl family (69 members) and the DOCK family (11 members) [2,3]. The RhoGDI family comprises the RhoGDIα, RhoGDIβ, and RhoGDIγ subtypes [4]. RhoGDIα is ubiquitously expressed, whereas RhoGDIβ is expressed mainly in hematopoietic cells and also in cancer cells. RhoGDIγ is expressed primarily in the brain. RhoGDIα and RhoGDIβ are expressed in phagocytes. The atypical subfamily of Rho-family GTPases is regulated by expression, post-transcriptional modifications, and/or interaction with other proteins but not by GEFs or GAPs [1,5]. Not all Rho-family GTPases are regulated by RhoGDIs [1,6].

Rho-family GTPases have various physiological functions in cytoskeletal regulation, cell polarity establishment, neuronal cell development, cell proliferation/division, cell movement/migration, cell-cell junction establishment, endosome trafficking, transcriptional regulation, reactive oxygen species (ROS) production, and tumorigenesis [41,42,43,44,45]. Rac, Cdc42, and RhoA are characterized both at the cellular (in vitro) and live animal (in vivo) levels. However, there is growing evidence for other Rho-family GTPases. Moreover, information about the functions of Rho-family GTPases in vivo is increasing and is based on various tissue (cell)-specific genetically engineered mouse models. In this review, we focus mainly on recent advances in research on Rho-family GTPases involved in highly polarized sensory organs/cells and ROS production as well as Rho-family GTPases related with cancer cells, which undergo uncontrolled growth and may be undifferentiated and/or disdifferentiated.

## 2. Hearing Function and Beyond

The organ of Corti (OC), the primary organ in the cochlea responsible for our sense of hearing, detects sounds by “electromechanical transduction (MET)” [46,47]. It has highly polarized sensory epithelial cells known as cochlear “hair cells (HCs)” and supporting cells (SCs) (Figure 1A). Cochlear HCs are arranged in a single row of inner HCs (IHCs) and three rows of outer HCs (OHCs). They have specialized actin-based structures such as stereocilia, apical junctional complexes (AJCs), and cuticular plates [48,49]. Sound-induced vibrations are detected by the directed and coordinated deflection of the stereocilia. Therefore, well-organized morphological and functional regulation is essential for the establishment and maintenance of hearing.

### 2.1. Role and Function of Rac in Hearing

The function of Rac1 in hearing was elucidated using *Foxg1*-Cre or *Pax2*-Cre mice with *Rac1* knockout (KO). The *Rac1*-KO (*Foxg1-Cre;Rac1^flox/flox^* and *Pax2-Cre;Rac1^flox/flox^*) mice showed embryonic lethality, incomplete development of HC planar cell polarity (PCP), and abnormal and fragmented hair bundles [50]. Mice with *Rac1* and *Rac3* double knockout (DKO) presented with exacerbated *Rac1*-KO phenotypes [51]. These pioneering studies unveiled the functions of Rac in the cochlea. However, Foxg1 [52,53] and Pax2 [54] are activated before Atoh1 in the cochlea, the master regulator of HC differentiation [55]. These genes are also active in the telencephalon (Foxg1), the brain stem, and throughout the OC epithelia including the HCs, SCs, and their precursors [56]. Further studies are required to reveal the specific functions of Rac in the HCs.

### 2.2. Role and Function of Cdc42 in Hearing

We recently reported that Cdc42 plays essential roles in the maintenance of cochlear HCs. We generated *Cdc42*-KO mice under the control of the *Atoh1* promoter (*Atoh1-Cre;Cdc42^flox/flox^*) [57]. After normal morphological maturation, the *Cdc42*-KO mice showed progressive sensorineural hearing loss (SNHL), particularly at high frequencies, and HC loss accompanied by various stereociliary abnormalities starting at postnatal day 14 (P14) (scattered, short, long, and fused) predominantly at the IHCs of the basal turn. Cdc42 acts on the membranes covering the stereocilia (especially the upper half) and the apical junctional complexes (AJCs) in the cochlear HCs [57]. Cdc42 functions at the AJCs as a complex with atypical subfamily of protein kinase Cs (aPKCs) [57]. However, its maintenance mechanism at the stereociliary membranes remains unknown. Active Cdc42 at the upper half of the stereociliary membranes may be involved in an “tip turnover” model of stereolilia, in which actin turnover in stereocilia occurs only at the tips, but not shafts [47,58,59,60]. The Cdc42 expression levels in the HCs resembled those in SCs [61]. Another group examined the Cdc42-KO phenotypes in *Fgfr3-iCre-ER^T2^;Cdc42^flox/flox^* mice (*Cdc42*-KO in SCs (Deiters’ and pillar cells)). Cdc42 was knocked out using *Fgfr3-iCre-ER^T2^* mice, in which Cre functioned with the assistance of tamoxifen administration (between P2 and P4 or P16 and P18). The mice presented with impaired apical polarization of SCs, no cochlear HC loss, and impaired wound healing in the SCs after ototoxin administration in the adults [62]. To examine the function of Cdc42 in HCs (especially OHCs), different tamoxifen administration intervals were applied at E13.5 and E14.5 or at E15.5 and E16.5 against *Fgfr3-iCre-ER^T2^;Cdc42^flox/flox^* mice. The mice showed impaired planar cell polarity (PCP) of the OHCs, fragmented stereociliary bundles, short stereocilia, and scattered OHC loss at P6 [63]. Their phenotypes were stronger than those observed in our *Cdc42*-KO mice [57]. While those studies used promoter-Cre mice, ours tested *Atoh1-Cre* transgenic mice. However, another group used *Fgfr3-iCre-ER^T2^* mice. Following are possible reasons why the phenotypes differed between these groups: (1) although the *Atoh1* and *Fgfr3* promoters function in both HCs and SCs, the *Atoh1* promoter operates predominantly in the former rather than the latter [56,64,65]; (2) unequal proportions of Cdc42 KO in the HCs and SCs may differentially affect the HC phenotypes since HC-SC interaction is a critical step in terminal HC differentiation [55].

### 2.3. Hearing in Patients with RAC1 or CDC42 Mutations

The physiological relevance of RAC and CDC42 has received increased attention. Patients have been identified with mutations in *CDC42*, *RAC1*, and *RAC2*. Takenouchi and Kosaki [29,30] and Motokawa et al. [31] reported three patients with a heterozygous missense point mutation in *CDC42*, which results in a p.Y64C mutant of CDC42. Their phenotypes included SNHL, macrothrombocytopenia (MTC), dysmorphic craniofacial features, lymphedema, and various CNS abnormalities such as ventriculomegaly and hypoplastic cerebellum (Figure 2). Martinelli et al. reported 15 patients from 13 unrelated families sharing nine a clinically heterogeneous but overlapping phenotype associated with heterozygous missense point mutations in *CDC42* mutations resulting in p.I21T, p.Y23C, p.Y64C, p.R66G, p.R68Q, p.C81F, p.83P, p.A159V, and p.Q171K mutants [32]. Patients with CDC42 mutants are now diagnosed with the Takenouchi-Kosaki syndrome (TKS) (Figure 2). Those with a heterozygous mutant (p.D57N) in RAC2 (a hematopoietic cell-specific RAC isoform) presented with recurrent infection and defective neutrophil functions [23,24]. Patients with a homozygous nonsense mutation in RAC2 (p.W56X) showed B-lymphocyte, T-lymphocyte, and neutrophil abnormalities [26]. In addition, p.C18Y, p.N39S, p.V51M, p.V51L, p.Y64D, p.P73L, and p.C157Y mutants of RAC1 were recently reported in seven patients [20]. They displayed various CNS abnormalities, including hypoplasia of the medial cerebellum and abnormal corpus callosum (Figure 2). Only one patient (p.Y64D) presented with SNHL. Hypoplasias of the medial cerebellum and corpus callosum are consistent with the results of previous studies using mice with DKO of *Rac1* and *Rac3* in the cerebellar granule neurons (*Atoh1-Cre;Rac1^flox/flox^;Rac3^−/−^*) [66], *Rac1* KO in the cerebellar neurons (*Nestin-Cre;Rac1^flox/flox^*) [67], and *Rac1* KO in the telencephalon (*Foxg1-Cre;Rac1^flox/flox^, Dlx6/6-Cre;Rac1^flox/flox^,* and *Emx1-Cre;Rac1^flox/flox^*) [68,69] (Figure 2). Rac1-KO in the telencephalon (*Foxg1-Cre;Rac1^flox/flox^*) was also associated with microcephaly [70]. However, the mechanisms by which heterozygous *CDC42* and RAC1 mutants induce the specific phenotypes remain unknown. Does each mutant have the same fundamental mechanism such as a gain-of-function or dominant negative effect? Is the dysfunction associated with each mutant compensated by intact molecules in the same subfamily (such as RHOJ and/or RHOQ for CDC42, and RAC2, RAC3, and/or RHOG for RAC1)? Further studies are required to answer these questions.

### 2.4. Deafness in Patients with Active DIA1 Mutations Downstream of RhoA

Fifteen formin proteins, which nucleate and elongate unbranched/straight actin filaments, are found in mammals and are classified into eight subfamilies [94,95]. DIA1 is one of three members (DIA1–DIA3) of the diaphanous-related formin (DRF) subfamily and is a downstream target of the RHOA signaling pathway [95]. DIA1 has two alternative splicing variants: DIA1-1 consists of 1263 amino acids (aa) and DIA1-2 has 1254 aa. In the latter case, in-frame exon 2 (27 nucleotide pairs; 9 aa) is skipped.

From the study of Cdc42 in cochlear HCs, we found that Cdc42 knockdown (KD) enhances RhoA signaling activation [57]. We hypothesized that the activation of DIA1 by RhoA at least partially accounts for the phenotypes observed in *Cdc42*-KO cochlear HCs. *DIA1* is the causative gene of the first type of non-syndromic SNHL with autosomal dominant inheritance, namely, DFNA1, which was reported in 1997 [96]. However, only one Costa Rican family was reported with this defect, and the molecular mechanism of DFNA1 was not disclosed. In 2016, we reported a novel heterozygous nonsense mutation in *DIA1-2* causing a constitutively active mutant of DIA1, namely, p.R1204X (p.R1213X in DIA1-1) [97]. The p.R1204X/R1213X mutation is located in the *C*-terminal diaphanous autoregulatory domain (DAD) and disrupts an inhibitory intramolecular interaction between the *N*-terminal diaphanous inhibitory domain (DID) and the DAD [97]. Patients with the p.R1204X mutation present with progressive SNHL, generally at high frequencies [97]. Transgenic mice with the p.R1204X mutation have a similar phenotype including progressive loss of cochlear HCs, particularly in the OHCs of the basal turn of the cochlea, progressive SNHL typically at high frequencies, various abnormal stereocilia (sparse, short, long, fused, and dislocated) of the HCs, and deformed cell-cell junctions with the SCs [97]. Other groups also reported DFNA1 caused by the constitutively active DIA1-1 mutants p.R1213X [98,99], p.R1210Serfs31X [100], p.R1210Glyfs31X [101], and E1192_Q1220del [101]. The first heterozygous mutation reported as a cause of DFNA1, p.Ala1221Valfs22X, is located outside the DAD (D1188-G1217 in DIA1-1) [96]. Nevertheless, all other mutants in *DIA1* manifesting as progressive SNHL are located in the DAD and produce constitutively active *DIA1* mutants. However, the molecular mechanism of the original DFNA1 mutant is still unclear.

### 2.5. Macrothrombocytopenia in Patients and Mice Associated with Rho-Family GTPases

DFNA1 is also accompanied by MTC and, occasionally, neutropenia [98,99]. Therefore, DFNA1 is a syndromic but never non-syndromic hereditary SNHL. MTC was also reported in *Cdc42*-KO (Figure 2), *RhoA*-KO, *Rock2*-KO, and *ADF/cofilin*-KO mice under the control of a megakaryocyte (MK)-specific promoter (using *Pf4*-Cre mice) [90,91,102,103,104,105]. The latter two mouse lines (*Rock2*-KO and *ADF/cofilin*-KO) have the downstream molecule KO in their RhoA signaling pathways. For *Rac1/Cdc42*-DKO (*Pf4-Cre;Rac1^flox/flox^;Cdc42^flox/flox^*) mice, *Rac1*-KO mice showed no MTC. However, the additional deletion of *Rac1* in *Cdc42*-KO mice markedly exacerbated their MTC phenotype [91]. The MTC in DFNA1 was explained by the reduction of proplatelet formation in cultured MKs and the increases in filamentous actin and stable microtubules in the platelets [98]. MTC in *Cdc42*-KO mice is theoretically compatible with that in DFNA1 because DIA1 is activated and enhanced in *Cdc42*-KD cells [57]. However, MTC in *RhoA*-KO [102], *Rock2*-KO [104], and *ADF/cofilin*-KO [103] mice contradicts our hypothesis that the activated RhoA signaling induces MTC. All patients with the CDC42 mutant p.Y64C presented with MTC even though the mutant was predicted to be active [32] (Figure 2). Non-muscle myosin heavy chain IIA (NMMHC-IIA or myosin heavy chain 9 (MYH9) encoded by the *MYH9* gene) is a downstream molecule of Rock-1 and Rock-2 [106]. Patients with *MYH9* mutations exhibit non-syndromic SNHL with autosomal dominant inheritance (DFNA17 (p.R705H) [107,108]) or various autosomal dominant syndromic disorders known as MYH9-related diseases (MYH9-RD). MYH9-RD (but not DFNA17) presents with MTC, which suggests that the symptoms associated with the *MYH9* mutation may differ among mutation sites [109,110]. MTC was also observed in MYH9-KO (*Pf4-Cre;MYH9^flox/flox^*) mice [111]. Taken together, these findings suggest that MTC is caused by disorganized (either decreased or activated) signaling in the cytoskeleton, including actin, tubulin, and myosin.

### 2.6. Deafness Associated with Rho-Family GTPases Other than Rac and Cdc42

Other Rho-family GTPases may also be involved in hearing. Mice with *RhoA* deletion in the OHCs and SCs (Deiters’ and pillar cells; *Fgfr3-iCre-ER^T2^;Cdc42^flox/flox^*) under tamoxifen administration at E13 and E14 showed only mild abnormalities of the developing OHCs (enlargement of apical cell surface) and extrusion of the OHCs in the endolymph. However, the PCP of their OHCs and SCs were normal [112]. *FAB65B* is the causative gene of DFNB104 and binds to RhoC at the tapering stereociliary base. Conventional *Fam65b*-KO mice presented with hearing loss as a consequence of morphological abnormalities at the base of the stereocilia [113]. HOMER2 belongs to the homer family of post-synaptic density scaffolding proteins. It binds to CDC42 through its CDC42-binding domain (CBD) in the CC domain. Patients with a p.R185P mutation in *HOMER2* showed hearing loss. Conventional *Homer2* KO-mice had progressive SNHL [114]. *ARHGFE6* is a specific Rho-GEF of RAC and CDC42 and one of the causative genes of X-linked mental retardation (XLMR). A reciprocal X/21 translocation presented with SNHL and severe intellectual disability [115]. Mice whose *Arhgef6* isoform 1 was lost by genome editing had stereociliary disorganization and progressive HC loss and SNHL [116].

### 2.7. Roles of Rho-Family GTPases in Other Sensory Organs

The Rho-family GTPases may also function in other highly polarized sensory cells/organs because they participate in polarization events [45]. For example, Cdc42 is involved in the development of the outer segment and the connecting cilia of retinal photoreceptor cells, which are modified and specialized primary cilia [79,80,117] (Figure 2).

## 3. Host Defenses through Superoxide Generation and Arrangements of Actin and Membranes

### 3.1. Superoxide Production from Rac-Dependent Nox2-Based Oxidase

Phagocyte NADPH oxidase (also known as gp91*^phox^* or Nox2) is heterodimerized with p22*^phox^* on the membranes and activated with the support of cytosolic p47*^phox^*, p67*^phox^*, p40*^phox^*, and Rac. Nox2 was the first and best characterized system identified to be under the regulation of the Rho-family GTPases [118] (Figure 1B). Nox2 activation is very tightly regulated [119,120,121,122,123,124,125]. Therefore, a genetic defect in any component of the Nox2 activation system results in the severe and recurrent infectious disease, chronic granulomatous disease (CGD) [123,126,127,128,129].

The first biological function ascribed to Rac was Nox2 activation [130]. The addition of either Rac1-GTP [131] or Rac2-GTP [132] was essential for high-level superoxide production in cell-free Nox2 assays. The role of Rac2 in vivo was later confirmed in hematopoietic cell-specific *Rac2*-KO mice [133] and the aforementioned RAC2 mutant patients [23,24,26]. Superoxide production from *Rac1*-KO neutrophils was normal [134]; however, that from DKO neutrophils of *Rac1* and *Rac2* had higher reducing power than that from Rac2-KO neutrophils [135]. Rac2 also participates in B lymphocyte activation and development [26,136] as well as those of T lymphocytes [25,26]. DKO of *Rac1* and *Rac2* almost completely suppress B lymphocyte development [78].

Phagocytosis is regulated by Rac, Cdc42, RhoA, RhoB, RhoC, and RhoG via actin and membrane rearrangements [137,138,139,140]. However, only the Rac subfamily (Rac1, Rac2, and Rac3) is essential for Nox2 activation [118,130]. RhoG, the fourth member of the Rac subfamily, may be involved in GPCR-mediated superoxide generation from neutrophils [141]. Nevertheless, a subsequent study revealed that RhoG is neither a direct activator nor a component of Nox2-based oxidase. Rather, it regulates superoxide generation by activating a Rac-GEF, namely, Dock2 [139,142].

Translocation (targeting) of Rac from the cytosol to the phagosomal membrane (phagosome) is independent of that for the ternary phox protein complex (p47*^phox^*-p67*^phox^*-p40*^phox^*) [143,144]. The latter is one of two cytosolic activator complexes in Nox2-based oxidase [123]. The separation of these mechanisms may account for the very tight regulation of Nox2-based oxidase. We reported the isoform-specific targeting mechanism of three Rac isoforms from the cytosol to the phagosome via their *C*-terminal polybasic (PB) motifs (KKRKRK in Rac1, RQQKRP in Rac2, and KKPGKK in Rac3) [120]. Rac1 directly targets the phagosome with the highly positively charged PB motif. In contrast, Rac2 initially targets the endomembranes which then fuse to the phagosome. Another study corroborated the delivery of Rac2 to phagosomes via the endomembranes (including granule/vesicle membranes) [145].

### 3.2. Superoxide/Reactive Oxygen Species (ROS) Production from Novel Noxs

Among the novel Nox isoforms (Nox1, Nox3, Nox4, Nox5, Duox1, and Duox2), three (Nox4, Duox1, and Duox2) are believed to provide hydrogen peroxide (H_2_O_2_) rather than superoxide for signaling. The superoxide produced by these three Noxs may be very quickly converted into H_2_O_2_ by an unproven mechanism [146,147,148,149,150,151]. We demonstrated that Rac1 helps activate Nox1, Nox2, and Nox3 [152] (Figure 1B). We also showed that the regulation of Nox1- and Nox3-based oxidases is less strict than that of Nox2-based oxidase [123,152,153]. Other groups also reported that Rac1 directly regulates Nox1 activation [154,155].

### 3.3. Regulation of Superoxide Production by RhoGDIs

The regulatory mechanism of the cytosol-membrane cycle of the RhoGDI-Rac complex is not fully understood. This complex is one of two cytosolic Nox2-based oxidase activators. It was proposed without evidence that the Rac dissociates from RhoGDI in the cytosol. We reported that the RhoGDIα-Rac1 and RhoGDIβ-Rac1 complexes translocate/target the phagosome then Rac1 dissociates from them on the membrane [125]. Another study also reported the dissociation of Rac from RhoGDIβ on the phagosome [156]. The dissociation mechanism has been most extensively studied in phagocytosis. However, our proposal may also apply not only to phagocytosis [139,157] but to other cell types and signaling pathways as well [4]. In general, the cytosol-membrane cycle of the RhoGDIα/β-RhoGTPase complex in which the RhoGTPases are activated (released from RhoGDIs and converted to GTP-bound form) on membranes may be regulated as follows: (1) interaction between the *C*-terminal PB motif in RhoGTPases and the negatively charged residues in the *N*-terminal of RhoGDIα/β to form a dimer complex in the cytosol [120,125], (2) anionic phospholipids such as PI(3,4,5)P_3_, PA, and phosphatidylserine produced on membranes [120,158,159] compete for binding to the PB motif in RhoGTPases with the *N*-terminal negatively charged residues in RhoGDIα/β, (3) RhoGTPases are released with the help of RhoGDI dissociation factors such as GEF and phosphorylation in RhoGDIα/β [160,161,162], thereby promoting their membrane insertion and activation, and (4) electric repulsion between the anionic membranes and the negatively charged residues in the *N*-terminal of RhoGDIα/β promotes the dissociation of RhoGDIα/β from the membranes [125].

## 4. Tumorigenesis

### 4.1. Recent Advances in Rac Involvement in Tumorigenesis

Rac1 is well known as a tumor progression factor (Table 1) because it participates in cell migration/invasion and proliferation [163]. Nevertheless, the exact mechanisms of Rac1 in tumorigenesis have not been fully unveiled. Recently, several activating RAC mutants including RAC1 (p.P29S, p.P29T, p.N92I, pC157Y, and p.A159V) and RAC2 (p.P29L and p.P29Q) indicated that RACs are oncogenic driver genes in human melanoma and head and neck squamous cell carcinoma patients and in cancer cell lines [14,15,21,22,27]. Although these studies reveal the roles of Rac in tumorigenesis and tumor progression, the mechanism of the involvement of Rac in tumorigenesis remains obscure. Using Forster resonance energy transfer (FRET) biosensors, Hirata et al. discovered that C6 glioma cells penetrating the brain parenchyma have high Rac1 and Cdc42 activity and low RhoA activity [164]. They also found that Dock9, a GEF for Cdc42, is an upstream mediator of this invasion process [164]. It was then demonstrated that Rac1 and Cdc42 activity is variable in C6 glioma [165]. Cells with high Rac1 or Cdc42 activity invade more efficiently and have stronger invasion-inducing signaling networks than cells with low Rac1 or Cdc42 activity [165]. Rac1b is an alternatively spliced Rac1 isoform with a short additional exon (3b; 57 nucleotides; 29 aa). Rac1b overexpression has been reported for breast, colon, and lung cancers [166]. In mouse lung and colon tumor models, Rac1b sufficed to initiate tumorigenesis but was assisted by inflammation or K-ras mutation [167,168].

### 4.2. A Novel Downstream Target of Rac Signaling, GSPT1, Is Associated with Tumorigenesis

We recently reported that the G1 phase of the primary astrocyte cell cycle is prolonged by both *Rac1*-KO and *Rac1*-KD. G1 to S phase transition 1 (GSPT1) is a novel transcriptionally regulated downstream target of Rac1 [169]. GSPT1 was initially identified as an essential gene for G1 to S phase transition in the cell cycle. However, it was later renamed as the eukaryotic releasing factor 3a (eRF3a) [170]. GSPT1/eRF3a is a GTP-binding protein participating in translation termination as a ternary complex with eRF1 and GTP. The eRF1 recognizes the terminal codon and releases the completed protein product after the hydrolysis of GTP by the intrinsic GTPase of eRF3a [170,171]. We found that Rac1-GSPT1 signaling promotes astrocyte cell cycle progression and proliferation (astrogliosis) after CNS damage such as spinal cord injury (SCI) and irradiation injury. The promotion of the G1 to S phase transition may be regulated by Rac1 via increases in cyclin D1 levels [172,173]. Genome-wide association studies (GWAS) showed that testicular germ cell tumors are susceptible to GSPT1 upregulation [174]. Xiao et al. also reported that HCT116 colorectal cancer cells express GSPT1 at high levels [175]. GSPT1 may be an antitumor drug target through ubiquitination and degradation [176]. GSPT1 may also be involved in lung and colorectal carcinoma cell migration [175,177]. In our studies, however, GSPT1 KD demonstrated no such effect in primary astrocytes, immortalized LN229 astrocytes, or A431 epidermoid carcinoma cells [169]. The reason for the discrepancy between our study and previous reports is unknown. Nevertheless, differences in the cell types and lines used may account for it. Another explanation is the difference in the cell migration assay used. A scratch assay is influenced by both cell migration and proliferation. Human GSPT1 is polymorphic and has a glycine repeat (7, 8, 9, 10, 11, or 12) in the *N*-terminus. The most common type is the 10 repeat [178]. Several studies reported that the presence of GSPT1 with a 12 glycine repeat is correlated with increased risks of gastric, breast, and colorectal cancers [179,180,181]. However, the exact mechanism linking the number of glycine repeats and tumorigenesis remains unknown. The RAC1-GSPT1 signaling axis may participate in an oncogenic mechanism and is a candidate target for a novel tumor therapy.

### 4.3. Involvement of Other Rho-Family GTPases in Tumorigenesis

Other members of the Rho-family GTPases including RhoA, RhoB, RhoC, Cdc42, RhoH, and RhoBTBs may also participate in tumorigenesis (Table 1) [1,14] (Figure 1C). A mutant in RHOA, p.G17V, causes a loss of wild-type RHOA function via the loss of GTP binding capability. It was reported that p.G17V drives cancer progression in 50–70% of all angioimmunoblastic T cell lymphoma (AITL) cases. AITL is one of the most common types of peripheral T cell lymphoma [8,9,10,182]. Other RHOA mutants such as p.G17del, p.G17E, p.C16R, p.T19I, p.D120Y, and p.A161E in AITL [8,9], p.R5Q (most frequent), p.Y42F, and p.Y42S in Burkitt’s lymphoma were also reported [11]. Activating mutations in *VAV1*, a GEF of RACs and RHOs [183], were reported as oncogenic drivers of peripheral T cell lymphomas [184,185,186,187]. The RHOA mutants, p.Y42C (most frequent), p.R5Q, and p.G17E, were reported in gastric cancer [12,13]. RhoA, RhoB and RhoC have oncogenic effects [1,14]. In addition, RhoB [17,18] and RhoBTBs [5] (mainly as RhoBTB2) may suppress tumors. RhoBTBs expression was reduced or silenced in various tumor types [5]. Patients with missense mutations in RHOBTB2 presented with epilepsy, severe intellectual disability, and postnatal microcephaly [39,188]. The function of RHOBTB2 in dendritic development was verified with a *Drosophila* model. Research on the tumorigenesis of RHOBTB2, RAC1, RAC2, RHOA, and CDC42 mutants in human patients is ongoing.

## 5. Conclusions

The functions of the Rho-family GTPases are gradually being elucidated, and innovative scientific and industrial applications are being developed for them. Studies using genetically engineered mice/animals (such as KO and knock-in) and information gathered from patients with genetic mutations are powerful tools to disclose the functions of Rho-family GTPases in vivo. Research on the Rho-family GTPases associated with sensory organs/cells involved in hearing, balance, and vision as well as those involved in various cancers, must be continued so that we gain a better understanding of the physiological role of these proteins in vivo. This will then facilitate the development of novel therapeutic modalities for sensory diseases and cancers.

## Figures and Tables

**Figure 1 cells-08-00092-f001:**
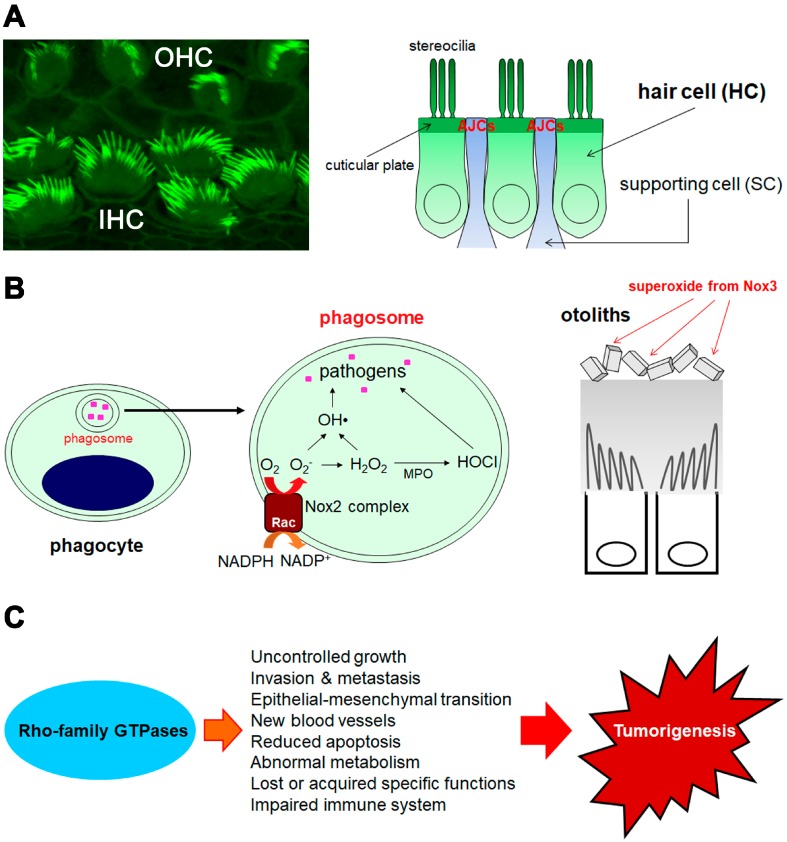
An overview of the main focus points of this review. (**A**) hearing function, (**B**) host defenses using ROS and Rac-dependent Noxs, and (**C**) tumorigenesis. (**A**) Inner hair cells (IHCs) and outer hair cells (OHCs) obtained from the wild-type cochlea at the age of P5 (stained by Alexa Fluor 488-labelled phalloidin). The illustration represents the mosaic alignment of HCs and supporting cells (SCs). AJCs: apical junctional complexes between HC and SC. (**B**) The illustration on the left shows phagocytes with a phagosome containing internalized pathogens (indicated by pink square). The middle illustration shows the magnified phagosome with a Nox2 complex (Rac is one of six components) on the membrane. Superoxide (O_2_^−^) and reactive oxygen species (ROS) are generated in the phagosome. The illustration on the right shows the otoliths on the otolithic membrane in the vestibule. Superoxide from Nox3 is essential for the synthesis of otolith; however, the origin of superoxide (that is Nox3-expressing cells in the inner ear) is still controversial. (**C**) The scheme represents how Rho-family GTPases are involved in tumorigenesis.

**Figure 2 cells-08-00092-f002:**
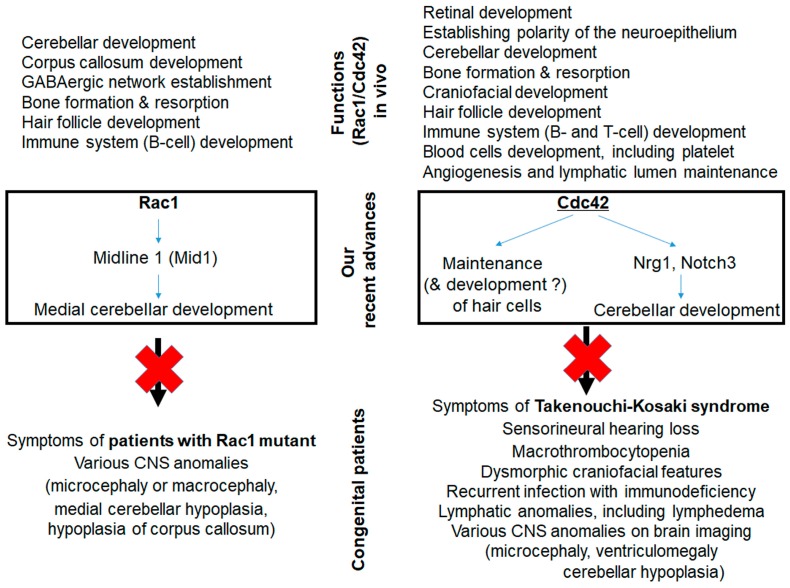
Functions of Rac1 (**left**) or Cdc42 (**right**) in vivo (**upper**), our recent advances using *Rac*- [66] or *Cdc42*-KO [57,71] mice (**middle**), and congenital/hereditary patients with RAC1 or CDC42 mutants (**lower**). Evidence from animal models helps elucidate the functions of Rac1 [66,67,68,69,72,73,74,75,76,77,78] and Cdc42 [71,74,79,80,81,82,83,84,85,86,87,88,89,90,91,92,93] and the symptoms of congenital/hereditary patients with mutations in RAC1 [20] and CDC42 [29,30,31,32]. Nrg1: neuregulin 1.

**Table 1 cells-08-00092-t001:** Knockout (KO) mice availability, congenital diseases, and tumorigenesis-associated Rho-family GTPases. Blue indicates typical (classical) Rho-family GTPases including the Rho, Rac, Cdc42, and RhoD/RhoF subfamilies. Red indicates atypical Rho-family GTPases including the Rnd, RhoH, RhoU/RhoV, and RhoBTB subfamilies. CNS: central nervous system, HD: host defenses, TKS: Takenouchi-Kosaki syndrome. +: conditional floxed mice. +^C^: conventional KO mice.

Name (Synonym)	KO Mice Availability	Congenital Diseases	Tumorigenesis
**Rho Subfamily**
RhoA	+ [7]		lymphomas [8,9,10,11], gastric cancer [12,13], head & neck squamous cell carcinoma [14,15]
RhoB	+^C^ [16]		induction [17] and suppression [17,18]
RhoC	+^C^ [19]		reported [14]
**Rac Subfamily**
Rac1	+	+ (CNS anomalies) [20]	melanoma [21,22], head & neck squamous cell carcinoma [14,15]
Rac2	+^C^	+ (HD deficiency) [23,24,25,26]	reported [14,27]
Rac3	+^C^		reported [14]
RhoG	+^C^ [28]		
**Cdc42 Subfamily**
Cdc42	+	+ (TKS) [29,30,31,32]	reported [14]
RhoJ (TCL)	+ [33]		
RhoQ (TC10)	+ [34]		
**RhoD/RhoF Subfamily**
RhoD			
RhoF (Rif)	+ [35]		
**Rnd Subfamily**
Rnd1			
Rnd2 (RhoN)			
Rnd3 (RhoE)			
**RhoH Subfamily**
RhoH	+^C^ [36]		lymphoma [37,38]
**RhoU/RhoV Subfamily**
RhoU			
RhoV			
**RhoBTB Subfamily**
RhoBTB1			
RhoBTB2		+ (CNS anomalies) [39]	suppression [5]
RhoBTB3	+^C^ [40]

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
