# Peer review of "Rho-Family Small GTPases: From Highly Polarized Sensory Neurons to Cancer Cells"

_cells, 2019, doi:10.3390/cells8020092_

Round 1
Reviewer 1 Report
The subject of this manuscript seems slightly too broad, especially in relation to the title. The title could be changed to better fit the content.
Re-writing the abstract by focusing only on relevant information will help the reader to better anticipate the nature of the article.
Concluding Remarks. I expect that the author can raise (or discuss) a number of key questions in the study of the Rho-family (current questions and future potential) in this section.
Overall, this is a very interesting and a comprehensive review of Rho-family small GTPases.
MINOR COMMENTS
The text should be properly proof read as Words are missing at spots in the text.
The reference number is duplicate, e.g. #90, 90; #180, 180, etc.
Author Response
1. Thank you very much for your helpful and thoughtful suggestions. If the reviewer accept my proposal, I would like to leave the title as it is, because the reviewer 3 recommends this title.
2. I have re-wrote the abstract for better understanding of this review article.
3. I have moved some sentences from the main text to the closing remark. In addition, I have modified the closing remark with an emphasis on the future potentials.
5. 6. I have re-checked and re-edited the whole of the manuscript, again.
7. The portions where I have corrected and modified are highlighted in yellow.
Reviewer 2 Report
The author has summarized the roles of Rho family GTPases in sensory neurons and cancer cells. The manuscript is well written and provide the readers with information and concepts interesting for the field.
I have only a major issue related to the part on tumorigenesis: in the Abstract section the author stated the review will focus on "inversely undifferentiated tumors". However, in part 4 (where involvement in tumorigenesis was presented) the author discussed the roles of these proteins and associated factors without direct mention of differentiation or related processes. I suggest modifying the sentence within the abstract since in the present version it does not summarize that elaborated in part 4.
Minor issues:
- Change "manipulated" with "engineered" in "genetically engineered mouse models" that has been used several times throughout the manuscript.
- In Table 1 what does mean the "+" alone symbol?
- In Table 1 all the reported roles in congenital disease or tumorigenesis have to be linked to specific references (this has been done only for a few examples). Please include the references.
- In the present form, Table 1 can be misleading for the reader in the column for tumorigenesis; it is not clear what gene was associated with the specific tumor type. Please align better the tumor types with the gene list on the left.
- Page 8: The author should be consistent in the way to refer to "G1 to S phase" with or without the hyphens.
- Line 285: The author should add the word "progression" after "cell cycle".
- Line 314: The word "extinguished" may not be the most appropriate.
Author Response
1. I used the word "inversely undifferentiated tumors" just as representing cancers, but not undifferentiated and most malignant type of tumors. Hence, to avoid misleading, I have removed the word "inversely undifferentiated." In addition, I have modified the abstract to focus on the topics described in this review article.
2. In accordance with the reviewer’s suggestion, I have changed "manipulated" with "engineered" in "genetically engineered mouse models" throughout the manuscript.
3. I have added the explanation of “+” in the Table 1 legend: it means conditional floxed mice.
4. In accordance with the reviewer’s suggestion, I have linked specific references to congenital disease and tumorigenesis in Table 1.
5. Thank you for thoughtful and useful suggestions. As pointed out by the reviewer, there are two different information in Table 1: tumor type and function (induction and/or suppression). However, it needs huge efforts to organize the genes according to the tumor-type. In addition, I believe that there is no strict correlation between the gene and the specific tumor-type. Hence, I have tried to clearly show the Table 1, as much as I can.
6. I have checked and used the words "G1 to S phase" throughout the manuscript.
7. I have added the word "progression" after "cell cycle."
8. The word "extinguished" have been changed with “silenced.”
7. The portions where I have corrected and modified are highlighted in yellow. I have one more round of English language editing.
Reviewer 3 Report
It is a review about the small GTPases of the Rho family, very well written, updated, interesting and relevant for the readers of Cells.The review is divided into three chapters: Hearing function and beyond, Host defenses through superoxide generation and arrangements of actin and membranes and Tumorigenesis. The author has titled the review in a very attractive way, "Rho-family small GTPases: from highly polarized sensory neurons to cancer cells" however the distribution in three blocks of isolated information makes the reader lose a point of interest. I think that the review would improve if the appropriate transitions between the chapters were made in order not to lose interest. And another point is that each chapter should present a schematic figure in order to favor the capture of information.
Author Response
Thank you so much for your helpful and thoughtful suggestions. In accordance with the reviewer’s suggestion, I have made a schematic figure (Figure 1). I believe that this figure will improve the transition of each chapter.